

# The antifungal activity of vapour phase of odourless thymol derivate

Róbert Kubinec[1], Jaroslav Blaško[1], Paulína Galbavá[1], Helena Jurdáková[1], Jana Sadecká[2], Domenico Pangallo[3], Mária Bučková[3] and Andrea Puškárová[3]

[1] Institute of Chemistry, Faculty of Natural Sciences, Comenius University in Bratislava, Bratislava, Slovak Republic
[2] National Agricultural and Food Centre—Food Research Institute, Bratislava, Slovak Republic
[3] Institute of Molecular Biology, Slovak Academy of Sciences, Bratislava, Slovak Republic

## ABSTRACT

Thymol is a substance with a great therapeutic potential possessing antibacterial and antifungal activity, with a characteristic odour that remains long after application but is not pleasant at higher concentrations. In this study, attention has been focused on describing the chemical and biological properties of the simply prepared trimethylsilyl ether of thymol (kubicin). Interestingly, kubicin has similar volatility as thymol, undergoes hydrolysis in the water (moisture; forming thymol and trimethylsilanol) and can be used at 6,000 times higher concentration than thymol without any negative and irritating odour. Kubicin showed diverse fungistatic and fungicidal activities when tested by direct contact assay, or in vapour phase. The volatile vapour of kubicin was effective on all tested fungal strains. These results suggest that vapours of kubicin might provide an alternative way to fight against fungal contamination.

## INTRODUCTION

The protection of cultural heritage is closely related to the protection of historical documents and artefacts, with an emphasis on their stability and conservation. From this point of view, moulds are a great enemy of archival and museum collections.

Moulds belong to the kingdom of Fungi and must survive by digesting organic materials for food (*Woodson, 2012*). Therefore, archive objects composed of cellulose-based and proteinaceous organic materials are potentially at risk (*NPS, 2007*). Moulds produce many tiny spores to reproduce which become airborne, travel to new location, and under the right conditions germinate (*NPS, 2007*; *NEDCC, 2012*; *Woodson, 2012*). This means that mould can permanently damage the materials supporting it, and make them more susceptible to future mould contamination (*NPS, 2007*; *Woodson, 2012*). It can stain wood, textiles and paper, and decrease the strength of their structures, making them more porous and fragile (*NPS, 2007*). As a by-product, mould can produce organic acids that will corrode and etch inorganic materials. Also, mould excretes pigments, metabolic products that color with age, digestive enzymes, odors, allergens, and toxins (*NPS, 2007*; *NEDCC, 2012*).

Corresponding author
Jaroslav Blaško,
jaroslav.blasko@uniba.sk

The most important factor in mould growth is the presence of moisture. The key of mould control is moisture control. Other factors that will contribute to mould growth in the presence of moisture are high temperatures, stagnant air, and the location of storage (*NPS, 2007*; *NEDCC, 2012*; *Woodson, 2012*).

Various methods have been used to prevent and eliminate fungal deterioration on paper-based materials. *Sequeira, Cabrita & Macedo (2012)* summarized the most widely used chemical and physical methods with their advantages and disadvantages in the review article. Physical methods, such as dehydration, gamma irradiation, high frequency current, UV radiation, freezing and others, use extreme temperatures, radiation or current, and do not required the application of chemical compounds. On the other hand, chemical methods are based on the application of chemical products, such as alcohols, ethylene oxide, formaldehyde, essential oils, thymol, pentachlorophenol, titanium dioxide, quaternary ammonium compounds and many others, also listed in mentioned review (*Sequeira, Cabrita & Macedo, 2012*). As physical methods leave no residue, they generally do not have a long-term effect. On the contrary, the prolongation of the antimicrobial effect when using chemicals is caused by persistent residues. For instance, *Michaelsen et al. (2013)* monitored various chemical and physical methods (ethylene oxide, freeze-drying and gamma rays) using fungi infected paper, and assessed their long-term efficacy to inhibit fungal growth according the molecular analysis (DNA and RNA recovery). The authors found that listed physical methods showed only short-term reduction in DNA and RNA recovery, which means that the mould can regenerate over time so these methods can only be used to remove mould to a controllable level (gamma rays) or to stop massive mould before further treatment (freeze-drying). Only the use of ethylene oxide fumigation proved negative for both DNA and RNA recovery and has long-lasting (one year) effect. Nevertheless, this chemical does not remain on treated materials, so the subsequent contamination is possible (*Michaelsen et al., 2013*).

However, the big disadvantage of chemical methods is that many of the most effective chemicals used in this field are toxic or carcinogenic (e.g., ethylene dioxide, formaldehyde, pentachlorophenol) (*IARC, 2006*; *IARC, 2008*; *IARC, 2019*) and are restricted in most of countries (*Sequeira, Cabrita & Macedo, 2012*; *Pietrzak et al., 2017*). In addition, the use of physical methods such as UV and gamma radiation can have undesirable side effects on treated paper in the form of reduced mechanical strength as well as yellowing (*Magaudda, 2004*; *Sequeira, Cabrita & Macedo, 2012*; *Pietrzak et al., 2017*; *Teixeira et al., 2018*). That is why developments in this area are still active. This is also evidenced by the recent publications (*Pietrzak et al., 2017*; *Teixeira et al., 2018*; *Karbowska-Berent et al., 2018*). One of the oldest agents used for disinfection since ancient Egypt is thymol or thyme oil (*Sequeira, Cabrita & Macedo, 2012*). During the 1950s pentachlorophenate and thymol were principal fungicides used to treat papyri (*Plenderleith & Werner, 1971*; *Cappitelli & Sorlini, 2005*), and thymol chambers have a long and traditional use to control fungus infections in cultural property in libraries and archives (*Jenkinson, 1937*; *Collis, 1970*; *Ellis, 1996*). Although in 1990s the thymol was identified as possible carcinogen (*Ellis, 1996*; *Isbell, 1997*), it has not been proven. On the contrary, thymol is a substance with a great therapeutic potential, and showed also anticancer properties

(*Nagoor Meeran et al., 2017*; *Salehi et al., 2018*). Moreover, thymol is listed by Food and Drug Administration as food additive (*EPA, 1993*; *FDA, 2019*), and is also proven that it does not represent an unacceptable risk to human health or the environment when is used as fungicide on table and wine grapes (*EFSA, 2012*) as well as to control mites on honey bees (*Health Canada PMRA, 2016*). Antibacterial and antifungal activities of thymol are summarized in the review by *Marchese et al. (2016)* with the conclusion that thymol possesses antibacterial and antifungal activity towards a large range of species encompassing biofilm-embedded microorganisms.

Thymol (5-methyl-2-isopropyl-1-phenol) is at room temperature in a colourless, transparent crystalline or flake form (*Isbell, 1997*). It has a characteristic odour that remains long after its application and is not pleasant at higher concentrations. Therefore, there was no massive use of thymol in the working environment.

Another described volatile compound with thymol skeleton is trimethylsilyl ether of thymol. An uncatalysed and $I_2$ catalysed silylation of phenols using hexamethyldisilazane under solvent free reaction conditions is published (*Jereb, 2012*). Trimethylsilyl derivatives of phenols were used for their gas chromatographic determination (*Betts, Allan & Donovan, 1984*), and they were also the subjects of interest for NMR spectrometry (*Jancke, Wolff & Lauterbach, 2010*). Recently, trimethylsilyl ether of thymol has been used as a substrate of dithiophosphorylation reaction under mild conditions (*Nizamov et al., 2016*). The characterization of trimethyl[5-methyl-2-(1-methylethyl)phenoxy]-silane; synonyms TMS-thymol derivate or thymol trimethylsilyl ether (kubicin) in terms of its physico-chemical properties is weak, and has not been described up to date in terms of sensory properties.

Usually, the odour level is measured by olfactometry; however, if a multicomponent mixture is measured, a separation step is required. Gas chromatography–olfactometry (GC–O) is an old technique and useful tool for investigating mixtures of odorous compounds (*Leland, 2001*). It consists of a gas chromatograph equipped with sniffing port beside the common detector. The column is split, and the separated components go to the detector and to the sniffing port at the same time, so the human nose can detect the peaks at the same time as the detector (*Benzo et al., 2007*).

The aim of this work was to prepare and describe the compound with comparable fungicidal activity, but significantly reduced odour, as thymol. The volatility of the new compound had to be comparable to that of thymol, in order to remove mould from paper and to mitigate their presence in air environment by using its vapour.

## MATERIALS & METHODS

### Synthesis of kubicin

To 10.0 g (66.5 mmol) of thymol (FCC, FG) was added approx. 3-fold excess of hexamethyldisilazane (HMDS) (ReagentPlus®, 99.9%); (15 g, 93 mmol), and 3 μL of trifluoroacetic acid (TFA) (for HPLC, ≥99.0%) as a synthesis catalyst (all purchased from Sigma-Aldrich, St. Louis, MO, USA). The reaction mixture was stirred at 50 °C for 30 min. Subsequently, the excess of HMDS, TFA and ammonia were evaporated to obtain

pure, colorless liquid kubicin (66.5 mmol, 14.7 g); $^1$H NMR (600 MHz, CDCl$_3$): $\delta$ 7.00 (d, $J = 7.8$ Hz, 1H), 6.68 (d, $J = 7.7$ Hz, 1H), 6.51 (s, 1H), 3.13 (hept, $J = 6.9$ Hz, 1H), 2.19 (s, 3H), 1.11 (d, $J = 6.9$ Hz, 6H), 0.21 (s, 9H); $^{13}$C NMR (151 MHz, CDCl$_3$): $\delta$ 152.58, 136.25, 136.18, 126.34, 122.29, 119.70, 26.91, 23.13, 21.24, 0.74. Purity of the product ($\geq$99.9%) was confirmed by gas chromatography—mass spectrometry (GC-MS) analysis in a full scan mode. The obtained mass spectrum was also compared with the mass spectrum (Fig. S1) of trimethyl[5-methyl-2-(1-methylethyl)phenoxy] silane from NIST database (NIST No. 108990) to structure confirmation. Moreover, UV–VIS and FTIR spectra were measured (see electronic supplementary material Figs. S2 and S3). All NMR experiments were recorded on Varian VNMRS 600 MHz spectrometer in 5 mm NMR tube. UV–VIS spectra were obtained using Agilent 8453 diode array spectrophotometer (Agilent Technologies, USA). ATR-FTIR analyses were performed using a Thermo Scientific Nicolet iS10 (Smart iTR diamond ATR) (Thermo Fisher Scientific, USA), and the sample was inserted directly onto the diamond surface of the optical cell.

## Chromatographic parameters
### The linear retention indices
The linear retention indices ($I_p$) of thymol and kubicin were calculated. Mixture of thymol, kubicin and C10–C24 $n$-alkanes (Supelco, Bellefonte, PA, USA), each 0.2 g/L in chloroform (Merck, Darmstadt, Germany), was separated by capillary gas chromatography. The programmed-temperature $I_p$ values were calculated on the number of carbons atoms (*Soják et al., 1973*; *Soják & Vigdergauz, 1993*) and confirmed by gas chromatography—mass spectrometry (GC-MS). GC-MS measurements were performed on a gas chromatograph 6890N Network GC System with a 5973 Network mass-selective detector (Agilent Technologies, Avondale, CA, USA). Separations of the mixture was carried out on columns: DB-5 30 m × 0.25 mm I.D. 0.25 µm coating, DB-23 60 m × 0.25 mm I.D. 0.25 µm coating, DB-WAX 60 m × 0.25 mm I.D. 0.25 µm coating, DB-17 60 m × 0.25 mm I.D. 0.25 µm coating, (purchased from Agilent Technologies, Avondale, CA USA ), and SPB-1 60 m × 0.25 mm I.D. 0.25 µm coating, (Supelco, Sigma-Aldrich, Germany). One µL of the mixture was injected in split mode (split ratio 1:20) at an injector temperature of 280 °C and a constant flow 1.5 mL/min of helium as the carrier gas. The program of temperature separation was from 50 °C (2 min) to 300 °C (2 min) with a temperature gradient of 10 °C/min. Mass spectral data were obtained in a Full Scan mode in the range of 33–450 amu. The transfer line temperature was 330 °C. Quadrupole conditions were as follows: electron energy 70 eV, and ion source temperature 230 °C.

### Boiling point determination
The boiling points were calculated using $I_p$ (*Soják et al., 1972*) obtained by GC-MS measurement with SPB-1 column for both, thymol and kubicin.

## Solubility in water
The solubility of thymol and kubicin was evaluated at two different temperatures 25 and 37 °C and was assessed as follows. Individual supersaturated solutions of the thymol and kubicin in the water (CHROMASOLV® Plus, for HPLC) (Honeywell, Seelze, Germany)

were prepared. Their concentrations in the saturated water phase were assessed using calibration curves prepared in acetone (AnalaR NORMAPUR® ACS, Reag. Ph. Eur.) (VWR, Rue Camot, France) within the range of 10–100 mg/L for kubicin, and 100–2,000 mg/L for thymol, measured by GC-MS. The number of points of calibration curves was 7. GC-MS conditions were as before (see chapter Chromatographic parameters), using DB-5 column, and MS detection was in SIM mode with selected $m/z$ 135 and 150 for thymol, and for kubicin 207 and 222 .

## Stability of kubicin in selected solvents

Kinetics of kubicin break-down and thymol formation at room temperature (25 °C) was measured by GC-MS (DB-5 column, SIM mode). Measurements were taken every 60 min within six hours, followed by two-hourly intervals up to 12 h, then after 16, 24 and 48 h. For stability test in water the saturated solution of kubicin was used. The concentration of kubicin in other solvents (methanol (for HPLC, gradient grade, ≥99.9%), DMSO (for spectroscopy Uvasol®), acetonitrile (for HPLC, gradient grade, ≥99.9%) purchased from Sigma-Aldrich, St. Louis, MO, USA and acetone) was 1% (v/v). Methanol and DMSO were used before and after drying using the molecular sieve 3 (Sigma-Aldrich, St. Louis, MO, USA).

## Gas chromatography—olfactometry analysis

GC-O analyses were performed using an Agilent 7890 GC equipped with a Gerstel olfactory detection port (ODP-3, Gerstel, Mühlheim, Germany). For GC-O experiments, effluent of the chromatographic column was split between a flame ionization detector (FID) and the sniffing port ODP with the split ratio 1:1. The samples were analyzed on a DB-WAX column (30 m × 0.32 mm × 0.25 µm, J&W Scientific, Folsom, CA, USA). The injection volume was 0.6 µL and the injector operated in splitless mode (0.5 min). The constant flow rate of the hydrogen carrier gas was 2.93 mL/min. The temperatures of both the injector and FID were 250 °C. The oven temperature program was as follows: the initial temperature of 80 °C was held for 1 min, then increased by 15 °C/min to 220 °C and held for 1 min. The temperature of the ODP transfer line heater was set at 230 °C. Added humidified air was pumped into the sniffing port at 40 mL/min to regenerate of nose mucosa. Each assessor of 3-member GC-O panel was properly trained to recognize odours by using solutions of artificial odorants. GC–O analysis was performed by injecting, at incremental dilutions, the solution of the thymol and kubicin in acetone, while an assessor signaled the perceived odours and their intensities by pushing a button, without seeing the chromatogram in progress. So, in parallel with a generating GC chromatogram was registered the respective aromagram. The sniffers also noted the perceived odour characteristics. At the same time, the computer recorded the retention time and sniffing time of compounds individually. The aroma intensity (AI) was evaluated using 6-point intensity scale from 0 to 6; 0 was none, 1 was threshold odour concentration, 4 was moderate, and 6 was extreme. The experiment was carried out in triplicate by each assessor. The incremental dilution of the sample proceeded by a factor 10, beginning from 10 mg/mL of each thymol and kubicin in acetone.

Results of GC-O analysis were expressed as average values of estimated aroma intensity in an intensity scale 0–6 with increments 1, obtained from 9 independent measurements for each sample, complying with the requirement of at least 8 citations within each olfactory percept. The value $\pm 1$ was considered as a standard error of estimation of odour intensities for applied intensity scale and engaged well trained sensory panel.

## Antifungal effect of kubicin
### Growth conditions and, solid diffusion assay
The antifungal activity of kubicin was assessed using different fungal strains (*Chaetomium globosum*, *Penicillium chrysogenum*, *Cladosporium cladosporioides*, *Alternaria alternata*, *Aspergillus fumigatus*, *Aureobasidium pullulans*, *Exophiala* sp.) isolated from museum environments and materials (*Kraková et al., 2012*; *Kraková et al., 2018*; *Benkovičová et al., 2019*). These fungal strains were cultivated at 25 °C on Malt Extract Agar (MEA).

The fungal suspension was prepared according to *De Lira Mota et al. (2012)*; final suspensions of conidia were adjusted using a Neubauer's chamber to $10^6$ conidia per mL.

The inhibition of fungal growth by kubicin was assessed using the disc-diffusion assay (direct contact) according the procedure described by *Puökárová et al. (2017)*. Briefly, the fungal suspension of each fungal strain (200 µL) was spread to MEA plates. On the surface of these MEA plates sterile paper discs were placed and 10 µL of kubicin was pipetted onto the discs. The discs impregnated with 10 mg of thymol diluted in DMSO were used as a positive control. A pure DMSO was included with each test to ensure that microbial growth was not inhibited by DMSO itself. Plates were incubated at 25 °C for 5 days. The diameter of the inhibition halos was measured in mm, including the diameter of the disc.

### Vapour diffusion assay
In order to demonstrate the fungistatic or fungicidal activity of volatile vapour of kubicin, small squares (5 × 5 mm) of the fungal mycelium grew on MEA were cut (using a steel borer) from the periphery of 7-day-old cultures and placed onto new MEA Petri dishes. Kubicin as the full-strength agent at dose levels of 1 and 2 µL/mL air space was placed on the inner surface of the Petri dish lid; the controls were not treated with kubicin. The Petri dishes were sealed with parafilm and were incubated inverted for 7 days at 25 °C. The fungal plugs with no extended mycelium were washed with sterile distilled water and reinoculated onto fresh MEA plates without adding kubicin. Transfer experiments and consequent confirmation of the fungistatic or fungicidal effect of kubicin vapours were carried out according to *Puökárová et al. (2017)*. If fungal strains start to growth after removal, there was a static effect, whereas if no growth occurs, the effect was fungicidal. Each experiment was made in triplicate. Effect of kubicin in the vapour phase on the test organisms is determined as the percentage of inhibition of the growth. The inhibition zone was measured in mm, without the square diameter. The percentage of inhibition means the percentage of growth for each individual microorganism compared with control treatment without kubicin, the diameter of squares was not included, % inhibition $= 100 - (T/C \times 100)$, where $T$ is colony diameter after the treatment with kubicin vapour and $C$ is control without exposition.

### Determination of fungistatic and/or fungicidal effect of kubicin vapour in model conditions

Model papers (35 mm × 20 mm, Whatman No. 1) were inoculated using 100 μL of the fungal suspension prepared as described above. The inoculated papers were directly placed on MEA plates. The paper models with 7-days-old cultures grown on it were placed in the reagent bottle and kubicin at different concentrations (0.002, 0.005, 0.01, 0.02, 0.04, 0.08, 0.12 μL/mL air space) were added on the inner surface of the bottle. Controls without kubicin were prepared. Each experiment was made in triplicate. After incubation at 25 °C for 7 days, the paper models that did not show any growth were transferred to the fresh MEA plates without adding kubicin for additional 7 days at 25 °C to check whether the protective effect was temporary or prolonged. If fungal strains on the model papers start to growth after transfer, there was a fungistatic effect, whereas if no growth occurs, the effect was fungicidal. The paper models, in which the fungal mycelial growth was not observed, were reinoculated into fresh Malt Extract Broth (MEB) without kubicin. No growth suggests a sporocidal ability.

The minimal inhibitory concentrations (MIC), minimal fungicidal concentrations (MFC) and minimal sporicidal concentration (MSC) of kubicin vapour are expressed as microliters of kubicin per volume unit of atmosphere above the organism growing in the reagent bottle that caused the fungistatic, fungicidal or sporocidal effect.

## Statistical analysis

The results represent a mean from 3 experiments ± standard deviation (SD). The differences between the given groups were tested for statistical significance using Student's $t$-test (*$p < 0.05$; **$p < 0.01$; ***$p < 0.001$).

## RESULTS

## Chromatographic parameters

Linear retention indices calculated using several capillary columns with different polarity of stationary phase for both thymol and kubicin are listed in electronic supplementary material (Table S1).

The boiling points of thymol and kubicin calculated from the measured retention indices using the non-polar columns were 232.0 °C (RSD 1.13%) and 237.4 °C (RSD 0.28%).

## Solubility and stability

The values of solubility at two different temperatures (25 and 37 °C) were 980 and 1,350 mg/L for thymol, and for kubicin 26 and 70 mg/L, respectively.

The dependence of kubicin concentration decrease in time with simultaneous increasing of thymol concentration in water and is shown in Fig. 1. Other time dependences in selected solvents are available in electronic supplementary material (Figs. S4–S9). The content changes of kubicin (%) with relative standard deviations from three measurements in the water and the other selected solvents at various times (1, 6, 12, 24 and 48 h) are listed in Table 1.

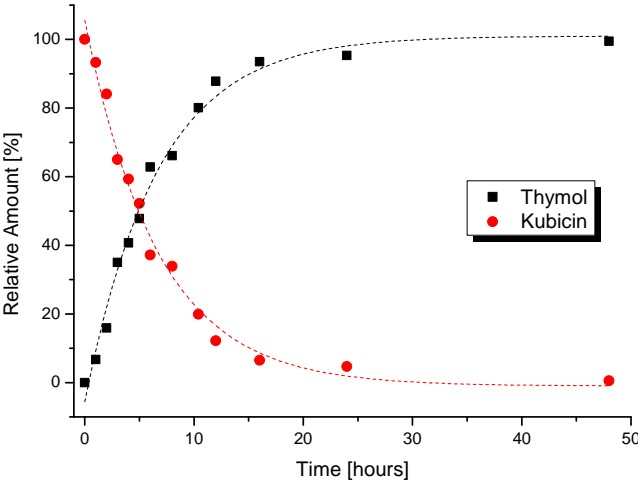

**Figure 1** **Stability time dependence of kubicin dissolved in water.**

**Table 1** **Kubicin content changes (%) with RSD values in selected solvents in dependence on the time.**

| Solvent | $t$ (hours) | | | | | |
|---|---|---|---|---|---|---|
| | 0 | 1 | 6 | 12 | 24 | 48 |
| Water | 100 (0.1) | 93 (0.5) | 37 (12) | 12 (5.8) | 4.4 (5.5) | 0 (2.4) |
| Methanol | 100 (0) | 93 (0.3) | 75 (0.3) | 61 (1.1) | 40 (0.6) | 17 (0.9) |
| Methanol[a] | 100 (0) | 85 (0.1) | 7 (6.6) | 0.1 (9.6) | 0.03 (7.0) | 0.01 (2.3) |
| DMSO | 100 (0) | 71 (1.3) | 31 (1.6) | 6.9 (1.3) | 1.0 (4.4) | 0.02 (11) |
| DMSO[a] | 100 (0) | 100 (0) | 98 (0.1) | 97 (0.1) | 94 (0.2) | 92 (0.3) |
| Acetonitrile | 100 (0) | 100 (0) | 100 (0) | 100 (0) | 100 (0) | 100 (0) |
| Acetone | 100 (0) | 100 (0) | 100 (0) | 100 (0) | 100 (0) | 100 (0) |

**Notes.**
[a] Dried solvents.

## Gas chromatography—olfactometry

The odour threshold concentration for kubicin (AI level 1) is concentration of 10 mg/mL in acetone, while for thymol this concentration represents AI level 6, extremely irritating (Table 2). After 100 times dilution of the prepared sample the AI level of thymol was 4, and it was possible to evaluate its odour characteristics as irritating, balsamic-thyme and cooling scent. Odour characteristic of kubicin at AI level 1 (none dilution) are as follows: neutral, nonspecific, and slightly different from background with slightly sweetish taste on the tongue. FID-chromatogram with aromagram of GC-O analysis is available in electronic supplementary material (Fig. S10).

The sample was diluted by factor 10 several times until AI level for thymol reached 0. Then the factor was adjusted to reach threshold odour concentration, and the final dilution was 6,000 times.

**Table 2 Determination of odour threshold concentrations (AI = 1) for kubicin and thymol perceived by way of GC-O.**

| Aroma-active compound | Aroma intensity (AI) | | | Odour quality |
| --- | --- | --- | --- | --- |
| | Concentration 10 mg mL$^{-1}$ ace | Concentration 0.1 mg mL$^{-1}$ ace | Concentration 1.66 µg mL$^{-1}$ ace | |
| Kubicin | 1 | Olfactorily undetected | Olfactorily undetected | Neutral, nonspecific smell; orthonasal olfaction leaves slightly sweetish taste on the tongue |
| Thymol | 6 | 4 | 1 | Irritating, balsamic-thyme and cooling smell |

Notes.
ace, acetone.

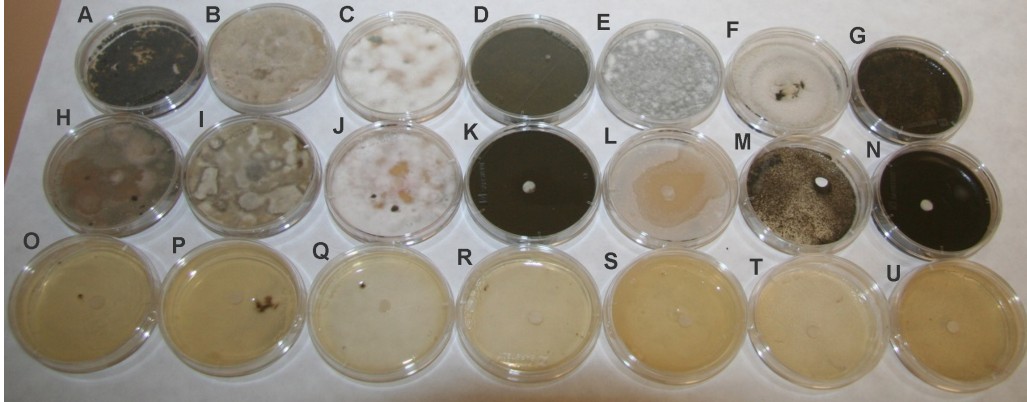

**Figure 2 Antifungal activity of kubicin and thymol against tested fungal strains.** (A, H, O) *Aspergillus fumigatus* (A, H, O), *Alternaria alternata* (B, I, P), *Chaetomium globosum* (C, J, Q), *Cladosporium cladosporioides* (D, K, R), *Penicillium chrysogenum* (E, L, S), *Aureobasidium pullulans* (F, M, T), *Exophiala* sp. (G, N, U). Control without treatment (A–G); treatment with kubicin (10 µL) (H–N); treatment with thymol (10 mg) (O–U).

## The antifungal activity

The in vitro antifungal activity of kubicin against fungal strains from environmental origin (*C. globosum, P. chrysogenum, C. cladosporioides, A. alternata, A. fumigatus, A. pullulans, Exophiala* sp.) was determined by the disc diffusion method by measuring the inhibition zones. The antifungal effectiveness of kubicin was depended on the microorganism species (Fig. 2). Kubicin appeared to have no antifungal effectiveness against *A. alternata, C. globosum, C. cladosporioides, Exophiala* sp. in direct contact solid assay. According to the results presented in Table 3, kubicin showed lower inhibition effect to the growth of *A. fumigatus, P. chrysogenum* with the inhibition zones ranging from 20–40 mm in comparison with thymol (inhibition of 88-90 mm, Petri dish diameter). Unclearly inhibition zones, which were difficult to measure were caused by kubicin against *A. pullulans*. The differences in the measured inhibition halos of kubicin on *A. fumigatus* ($p = 0.002$) and *P. chrysogenum* ($p = 0.0014$) were statistically significant from the control. These two fungal strains were highly sensitive (**$p < 0.01$). Thymol showed antifungal effects (***$p < 0.001$) against all tested strains (Table 3).

**Table 3** **Antifungal activities of kubicin and thymol, solid diffusion test: total inhibition (mm) including disc diameter.** The effects of kubicin (10 μl) and thymol (10 mg, dissolved in DMSO) are shown. Pure DMSO was also used, no inhibition was observed. Results are mean ± SD values of three independent experiments.

| Fungal strains | Kubicin | Thymol | DMSO |
|---|---|---|---|
| *Aspergillus fumigatus* | 35,21 ± 3,00[**] | 89,33 ± 0,58[***] | 0 |
| *Alternaria alternata* | 0 | 89,66 ± 0,58[***] | 0 |
| *Chaetomium globosum* | 0 | 89,00 ± 1,00[***] | 0 |
| *Cladosporium cladosporioides* | 0 | 88,60 ± 1,53[***] | 0 |
| *Penicillium chrysogenum* | 40,00 ± 2,65[**] | 88,30 ± 1,16[***] | 0 |
| *Aureobasidium pullulans* | 0 | 88,66 ± 1,53[***] | 0 |
| *Exophila sp.* | 0 | 89,30 ± 1,16[***] | 0 |

Notes.
[**]$p < 0.01$.
[***]$p < 0.001$ indicate statistically significant differences compared to each control treatment without test substances (Student's $t$-test). 0 mm represents no inhibition.

For further kubicin activity evaluation the inhibition of fungal growth after treatment with kubicin vapour was tested by diffusion assay. Mycelial growth inhibition of kubicin vapour against selected fungal strains on Malt Extract Agar plates is shown in Fig. 3. Inhibition of fungal growth in the vapour phase was the most effective (76.2–99.9%, *** $p < 0.001$) in comparison with the solid diffusion test. Kubicin (1 μL/mL air space) in vapour phase inhibited the mycelial growth of all tested fungal strains (Fig. 3) and in case of *A. pullulans* the inhibition of mycelial growth was confirmed as fungicidal effect. It was observed completely inhibition (99.9%). For this fungal strain kubicin vapour gave a stronger inhibition of growth, than kubicin in direct solid assay (unclearly inhibition zone). Similar results were obtained for all studied fungal strains, as confirmed by statistical analyze (Fig. 4). Kubicin at concentration of 2 μL/mL air space gave a strong inhibition. Both concentration levels induced a significantly different level of inhibition than those observed in the untreated control strains (*** $p < 0.001$). The re-inoculation of inhibited fungal mycelial plugs for which no growth was observed into fresh MEA plates confirmed a fungicidal effect of kubicin.

MIC, MFC and MSC of kubicin vapour were performed in model conditions using inoculated paper samples in the bottles (Fig. 5A). The results obtained from these assays are summarized in Table 4. These *in vitro* trials permitted to find which amounts of concentration of kubicin should be used in order to have a reliable disinfection result against environmental fungi. The volatile vapour of kubicin in model conditions was effective on all tested fungi except *Exophiala* sp. (Table 4). The tested substance kubicin (0.005 μL/mL air space) completely inhibited the mycelial growth and showed also fungicidal effect on inoculated paper samples with *P. chrysogenum, C. cladosporioides, A. alternata, A. fumigatus, A. pullulans,* (Figs. 5B–5K). For *C. globosum* kubicin vapour gave fungicidal effect (with MFC of 0.002 μL/mL air space). The sporocidal effect was confirmed after re-inoculation into fresh malt extract broth. The sporocidal effect was confirmed for *C. globosum* (with MSC of 0.002 μL/mL air space), *C. cladosporioides* and *A. alternate* (with MSC of 0.005 μL/mL air space) (Figs. 5L–5M).
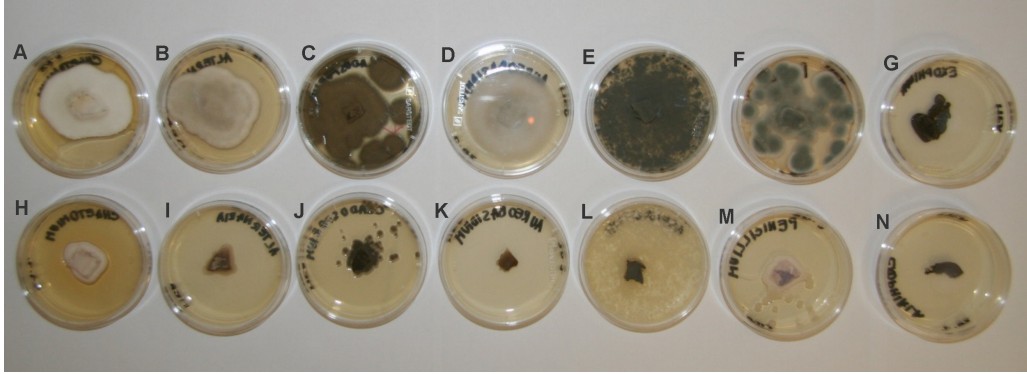

**Figure 3 Mycelial growth inhibition of kubicin vapour against selected fungal strains on Malt Extract Agar plates.** (*Chaetomium globosum* (A, H), *Alternaria alternata* (B, I), *Cladosporium cladosporioides* (C, J), *Aureobasidium pullulans* (D, K), *Aspergillus fumigatus* (E, L), *Penicillium chrysogenum* (F, M), *Exophiala* sp. (G, N). Control without treatment (A–G); the inhibition of fungal growth after treatment with kubicin (1 µL/mL air space) (H–N).

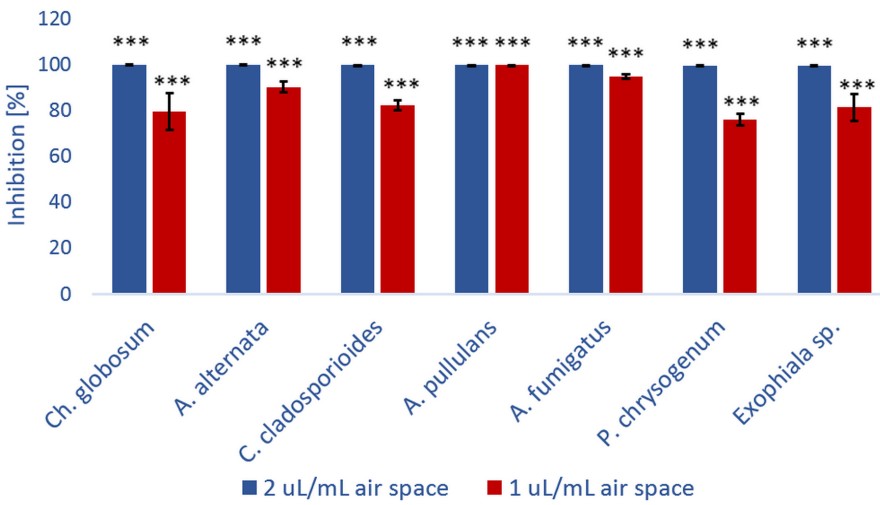

**Figure 4 The percentage of growth inhibition of fungal strains (%) versus kubicin vapour concentration (µL/mL air space).** Results are mean ± SD values of three independent experiments. *** $p < 0.001$ indicate statistically significant differences compared to each control treatment without test substances (Student's *t*-test).The values shown in the table were confirmed in three independent experiments in which the effect of kubicin on the growth of fungal isolates was monitored. The aim of the experiment was the determination of fungistatic and/or fungicidal effect of vapour phase of kubicin in model conditions in the presence of the test substance using these concentrations: 0.002, 0.005, 0.01, 0.02, 0.04, 0.08, 0.12 µL/mL air space. The physiological effect of the test substance (kubicin) was compared to control isolates (fungi) growing in the absence of the test substance. The same effect with the same concentration (specific for a given fungus) was observed during three different experiments.

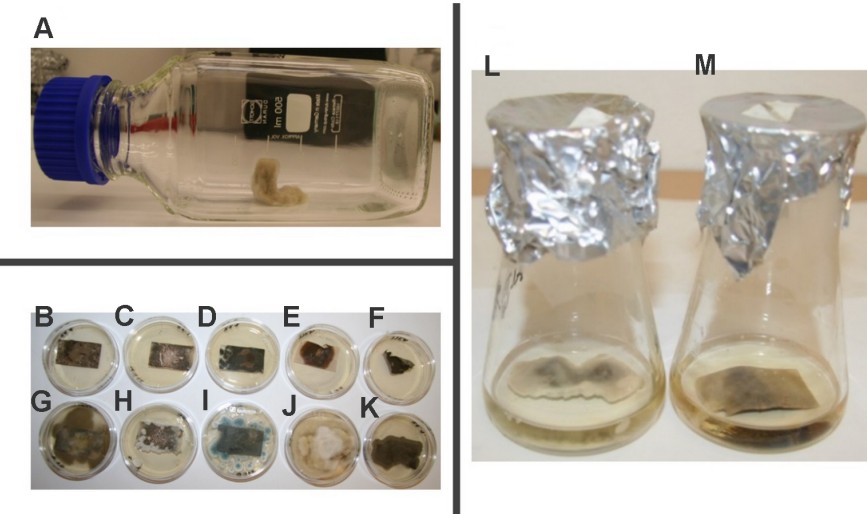

**Figure 5 The effect of kubicin vapour on contaminated model papers.** The bottle system with contaminated model paper (A). The example of confirmation of the fungicidal (B–F) and fungistatic (G–K) effect on inoculated paper samples after re-inoculation onto fresh agar medium. *Aureobasidium pullulans* (B, G), *Penicilium chrysogenum* (C, H), *Aspergillus fumigatus* (D, I), *Alternaria alternata* (E, J), *Cladosporium cladosporioides* (F, K). The revival of the growth of *Alternaria alternate* after re-inoculation into fresh maltextract broth (L) and the sporicidal effect of kubicin against *Alternaria alternata* (M).

**Table 4 Minimal inhibitory concentrations (MIC), minimal fungicidal concentrations (MFC) and minimal sporicidal concentrations (MSC) ($\mu$L/mL air space) of kubicin derivate against the tested fungal strains.**

| Fungi | MIC | MFC | MSC |
|---|---|---|---|
| *C. globosum* | 0.002 | 0.002 | 0.002 |
| *A. pullulans* | 0.002 | 0.005 | NE |
| *A. alternata* | 0.002 | 0.005 | 0.005 |
| *P. chrysogenum* | 0.002 | 0.005 | NE |
| *C. cladosporioides* | 0.002 | 0.005 | 0.005 |
| *A. fumigatus* | 0.005 | 0.005 | NE |
| *Exophiala* sp. | NE | NE | NE |

**Notes.**
NE, no effect.
The concentrations were calculated in microliters of kubicin per mililiter of the atmosphere sampled.

## DISCUSSION

Very close linear retention indices for the non-polar SPB-1 and DB-5 columns clearly indicate that the prepared kubicin has similar volatility as thymol. $I_p$ values for both compounds and polar columns demonstrate the possibility of the reliable use of these columns in the assessment of sensory properties (due to the sufficient separation of thymol and kubicin) without affecting the odour assessors. The measured $I_p$ values for thymol using nonpolar columns are identical with the literature values (*Benzo et al., 2007*). The calculated boiling point of thymol using $I_p$ is in accordance with the published value of

233 °C (*Haynes, 2014*). The value of kubicin boiling point was not published until now. For the columns with higher polarity, the calculated values are different from real values due to the contribution of other interactions in addition to dispersion forces.

The significantly higher solubility of thymol compared to kubicin results from the presence of free OH group. The solubility of kubicin in water increases significantly at higher temperature.

Kubicin undergoes hydrolysis in the water (Fig. 1; forming thymol and trimethylsilanol), as well as in solvents containing small content of water. In the case of methanol, kubicin rapidly decomposes to form thymol and methoxytrimethylsilane. This kubicin decomposition in the presence of water and/or methanol also results in an unpleasant irritating characteristic odour of thymol.

The neat kubicin is almost odourless. Interestingly, the olfactomery experiments show that kubicin can be used at 6,000 times higher concentration than thymol without any negative and irritating odour.

Since the most important factor for fungal growth is the presence of moisture, we assume that the mechanism of antifungal action of kubicin is based on the hydrolysis of kubicin during its contact with the mould, which leads to the formation of thymol, whose antimicrobial effect and possible mechanism of action are well known and described in literature (*Trombetta et al., 2005*; *Marchese et al., 2016*). After the hydrolysis, produced thymol can directly interact with the microorganism plasma membrane, resulting in disruption of membrane integrity and in leakage of intracellular materials. Moreover, thymol can penetrate the cell and interact with intracellular lipid membranes and thus results in cell death (*Trombetta et al., 2005*). Thymol also induced conidial apoptosis in Aspergillus flavus via stimulating $K^+$ eruption (*Hu et al., 2018*).

In the first series of antifungal assays we used a direct contact. The measured inhibition zones of kubicin and thymol indicated that thymol was more effective than kubicin in solid diffusion. The strong antifungal activity of thymol in our study is in accordance with previous finding (*Marchese et al., 2016*). Our results revealed that kubicin was more effective in vapour state against tested fungal strains, both on solid media and surface of model papers in atmosphere generated by kubicin.

Thymol and thyme oil have been used as fungicides in the protection of the cultural heritage from ancient times to the present, and their fungicidal effects have been documented in many studies. In a recently published study, *Pietrzak et al. (2017)* investigated the effectiveness of three different methods (low temperature plasma, thyme essential oil microatmosphere and silver nanoparticles misting) for disinfecting microbially contaminated history books. Low temperature plasma and thyme essential oil were more effective compared to the silver nanoparticles misting. Moreover, thyme essential oil showed a wider scale of fungicidal activity compared to the other two methods. It should be noted that each of the three methods has no significant effect or has a positive effect on the treated paper in terms of its optical and mechanical properties. Contrary to these methods, the tea tree oil and hydrogen peroxide showed unacceptable side effects (*Karbowska-Berent et al., 2018*), moreover, due to hydrogen peroxide is an oxidizing agent, its usage is always controversial in this field. Even ethanol exhibits some side effects (loss

of gloss, dehydration of cellulose fibers) but the application of its concentration lower than 70% allows reducing some of them. In addition to the aforementioned advantages of thymol-based agents, further indisputable advantage of these compounds is the financial and instrumental modesty compared to methods such as the low-temperature plasma (*Pietrzak et al., 2017*) or the application of supercritical fluid carbon dioxide (*Teixeira et al., 2018*). However, the disadvantage of thymol application is the irritating odour, even in low concentration levels, which requires the usage of special room or chamber. As shown in this paper, this problem can be overcome by using kubicin which can be applicated even at several thousand higher concentrations than thymol. Although we suppose that kubicin mode of action as a fungicidal agent results in the formation of thymol, and it does not produce an unpleasant characteristic odor due to further interaction of thymol with the mould cells. Therefore, the vapour of kubicin represent an antifungal agent that can be applied as currently available disinfection method. This study gives kubicin great potential to become a candidate for the development of alternative methods to control environmental undesirable fungi. However, the use of kubicin in larger real space requires further research due to the fact that up to several hundreds of milliliters of pure substance, depending on the size of the room (e.g., 250 mL of kubicin/50 cubic meters), need to be evaporated and homogenously distributed throughout the space to achieve desired antifungal effect. At the beginning, this type of room could have small dimensions and could be used for the disinfection of single objects in various restoration and conservation laboratories. Additional research related to its effect on different historical objects (papers, prints, photographs) is needed as well.

## CONCLUSIONS

The odourless trimethylsilyl ether of thymol (kubicin) was synthesized. The prepared compound is liquid, has comparable volatility as thymol, and slowly hydrolyses to thymol and trimethylsilanol in water; however, leaves no unpleasant odour in the working environment. Thanks to its characteristics, kubicin can be used in several applications that allow the inhibition or mitigation of the fungal load. Kubicin vapour can be employed for the disinfection, for example, of various archival documents, by the development of a closed system where objects can be inserted in and kubicin will diffuse inside too. Kubicin can also be spread into deposits or exhibition rooms (archives, libraries and museums) in order to reduce the presence of airborne mycobiota of these environments.

### Funding

This publication is the result of the implementation of the projects ITMS 26240120025 and 26240220071 supported by the Research & Development Operational Programme which is funded by the ERDF. Work was also supported by the Slovak Research and Development Agency under the contract numbers APVV-15-0466 and APVV-18-0282 and by VEGA

Agency with the projects 2/0059/19 and 2/0061/17. The funders had no role in study design, data collection and analysis, decision to publish, or preparation of the manuscript.

**Grant Disclosures**

The following grant information was disclosed by the authors:
Research & Development Operational Programme which is funded by the ERDF: ITMS 26240120025, 26240220071.
Slovak Research and Development Agency: APVV-15-0466, APVV-18-0282.
VEGA Agency with the projects: 2/0059/19, 2/0061/17.

**Competing Interests**

The authors declare there are no competing interests.

**Author Contributions**

- Róbert Kubinec conceived and designed the experiments, authored or reviewed drafts of the paper, and approved the final draft.
- Jaroslav Blaško performed the experiments, analyzed the data, prepared figures and/or tables, authored or reviewed drafts of the paper, and approved the final draft.
- Paulína Galbavá and Mária Bučková and Andrea Puškárová performed the experiments, prepared figures and/or tables, and approved the final draft.
- Helena Jurdáková analyzed the data, authored or reviewed drafts of the paper, and approved the final draft.
- Jana Sadecká performed the experiments, authored or reviewed drafts of the paper, and approved the final draft.
- Domenico Pangallo conceived and designed the experiments, analyzed the data, authored or reviewed drafts of the paper, and approved the final draft.

**Data Availability**

   The raw data are available in the Supplemental Files.

**Supplemental Information**

Supplemental information for this article can be found online at http://dx.doi.org/10.7717/peerj.9601#supplemental-information.

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
