# Peer review of "The antifungal activity of vapour phase of odourless thymol derivate"

_PeerJ, doi:10.7717/peerj.9601_

## Round 0.1 · original submission · Major Revisions

After reading reviewers comments, I think your manuscript would be greatly improved if you should incorporate into your manuscript the different comments made by the reviewers. Mainly, reviewers are concerned about the poor organization and presentation of the manuscript, English usage, and poor description of your experimental procedures, and lack of statistical analysis. After reviewing and re-organizing your manuscript, it will be sent for a second round of reviews, before it can be considered again for publication. Please try to attend reviewers requests to the best of your ability and feel free to contact me if you have any questions before you re-send the text.

Reviewer 1 ·

Basic reporting

There is a lack of statistical analysis of the experimental results.

Experimental design

no comment

Validity of the findings

no comments

Additional comments

The paper suggests a new idea and material to fight against fungal contamination to protect cultural heritage.

I review the manuscript. The paper suggests a new idea about using the vapour of kubicin to fight against fungal contamination to protect cultural heritage. However, I would like to comment on some parts that could increase the readability of the paper.
1. Page 6, line 67, “The authors conclude that only the use of 68 ethylene oxide fumigation has long-lasting (one year) effect, but this chemical does not remain 69 on treated materials, so the subsequent contamination is possible”. The sentence is inconsistent
2. Line 139, “To” was removed
3. There is a lack of statistical analysis of the experimental results. No variability parameter (standard deviation, standard error, etc.) was included in tables. Hope you suppling these data.
4. In fig 2 and 3, you need to mark a, b, c on the fig and note what they are for every strain under the fig.
5. In the part of Antifungal effect of kubicin. Is kubicin dissolved in water or in an organic solvent? if it was dissolved in an organic solvent, is there a control group with only added organic solvents?

Reviewer 2 ·

Basic reporting

- English is so poor. English should be reviewed by native speakers.

- Introduction: Too long. It is necessary to shorten the Introduction section.

- Discussion: Too short and few references. No deep discussion about their finding. No discussion about prior studies.

Experimental design

1. Materials & Methods
- All chemicals: Purity should be described.
- Synthesis of kubicin: Mass data?
- Antifungal activity: Only figure (Fig. 2 and Fig. 3) is shown. Where is data?
It is necessary to describe fungicidal activity as % or length of inhibition
zone. No statistical analysis. Only one concentration of kubicin was
tested for contact and vapor phase antifungal activity (FIg. 2 and Fig. 3)
- Data for GC-O analysis is only described as text. No figure or table. It is necessary to describe mean and standard deviation (or error).
- Table 1: Only one replication? Describe as mean and standard deviation (or error).
- Table 2: what is unit? uL/mL air?

Validity of the findings

- Interesting finding but lack of data and poor presentation of their result.

Additional comments

This manuscript is about the antifungal activity of kubicin.

Experimental design and presentation of results are not logic.

This manuscript should be redesigned and rewritten for publication.

---

## Round 0.2 · Minor Revisions

Dear authors, Let me start by apologizing for the delay in this response, which was mainly due to my unsuccessful wait for other reviewers to respond to my call. Before accepting this article for publication, and after having read the manuscript myself, I would like you to discuss the practical implications of your findings. Mainly, how a librarian should/could proceed in order to use these findings? Do you think that a description of a brief step by step recipe could be added to the text? If you were a librarian, how would you use your findings? In my opinion, this description would not only simplify the use of these findings, but also help people who would want to use this technology. Please let me know if this would be possible. After I receive your response, I will make an acceptance determination as soon as possible.

Other comments.
Figure 1. Please change: Relative amount in % to Relative Amount [%]
Figure 4. % inhibition to Inhibition [%]

Reviewer 1 ·

Basic reporting

no comment

Experimental design

no comment

Validity of the findings

no comment

Additional comments

Fig 4 % inhibition change to inhibition (%)

---

## Round 0.3 · accepted · Accept

I am glad to report to you that your article has been accepted and I thank you for your patience and willingness to incorporate and assimilate suggestions. I would really like to see this work expanded to and tested in real-life applications.